# Assessment of Food Intake Assisted by Photography in Older People Living in a Nursing Home: Maintenance over Time and Performance for Diagnosis of Malnutrition

**DOI:** 10.3390/nu15030646

**Published:** 2023-01-27

**Authors:** Anne Billeret, Chloé Rousseau, Rémy Thirion, Béatrice Baillard-Cosme, Kevin Charras, Dominique Somme, Ronan Thibault

**Affiliations:** 1CHU Rennes, Service d’Endocrinologie-Diabétologie-Nutrition, Centre labellisé de Nutrition Parentérale au Domicile, INRAE, INSERM, Univ Rennes, NuMeCan, Nutrition Metabolisms Cancer, F-35000 Rennes, France; 2CIC 1414, INSERM, CHU Rennes, F-35000 Rennes, France; 3CHU Rennes, Service de Gériatrie, F-35000 Rennes, France; 4CHU Rennes, Living Lab Vieillissement et Vulnérabilités, F-35000 Rennes, France; 5CHU Rennes, Service de Gériatrie, CNRS, Arènes—UMR 6051, Inserm, RSMS—U 1309, Living Lab Vieillissement et Vulnérabilités, Univ Rennes, F-35000 Rennes, France

**Keywords:** undernutrition, food intake, muscle, sarcopenia, body composition

## Abstract

Malnutrition is related to poor outcomes. Food intake semi-quantitative assessment is helpful for malnutrition screening. Aims: to assess maintenance over one month of one-day semi-quantitative assessment of food intake (primary aim) and its performance in diagnosing malnutrition (secondary aim) in older people living in a nursing home. Food portions consumed at lunch and dinner were measured during 20 days by the Simple Evaluation of Food Intake (SEFI)^®^ assisted by photography (SEFI^®^-AP) in 70 residents. Nutritional status was assessed in each patient during the first week of food intake monitoring according to Global Leadership Initiative on Malnutrition criteria. Food intake was decreased, i.e., SEFI^®^-AP < 7, in 39% (n = 27/73) of patients. According to the methods, 36 to 48% (n = 25 to 33/73) of patients had malnutrition, and 6 to 37% (n = 4 to 25/73) sarcopenia. According to a generalized linear model on ranks with repeated measures, the SEFI^®^-AP medians of lunch (*p* = 0.11) and means of lunch and dinner (*p* = 0.15) did not vary over time. Day 3 SEFI^®^-AP anticipated decreased food intake from days 4 to 20, with a sensitivity of 78% (95% confidence interval (CI), 62–94), a specificity of 30% [95%CI, 17–44] and positive and negative predictive values of 41% [95%CI, 28–55] and 68% [95%CI, 48–89]. The performance of SEFI^®^-AP for diagnosis of malnutrition using calf circumference <31 cm as a phenotypic criterion was correct: area under the curve = 0.71 [95%CI, 0.59–0.83]. SEFI^®^-AP sensitivity was better if ≤9.5 than <7, and inversely for specificity. The food intake of older people living in nursing homes is stable over one month. One-day SEFI^®^-AP correctly anticipates food intake during the following month and predicts diagnosis of malnutrition. Any decrease in food intake should lead to suspect malnutrition.

## 1. Introduction

Malnutrition is common and worsens the clinical outcome of older people living in nursing homes [1]. Weight loss of more than 5% in one year and a body mass index (BMI) of less than 20 are associated with higher 6-month mortality [2]. Nutritional support improves nutritional status and reduces risk of complications and loss of autonomy [3]. Lack of systematic identification and management of malnutrition decrease chance of survival, health and quality of life [4]. Sarcopenia is the most common geriatric syndrome [5] and is associated with higher mortality. Management of malnutrition and sarcopenia is, therefore, a major challenge among older people living in nursing homes [1,6].

Decreased food intake is one of the main etiologies of malnutrition in older people, mainly because of age-related anorexia [7]. At hospital, eating less than 50% of the food served at lunchtime is associated with doubled mortality at one month [8]. If malnutrition was earlier and better diagnosed in nursing homes, dedicated nutritional care would be triggered earlier, avoiding malnutrition becoming chronic and worsening.

New criteria for diagnosis of malnutrition in people aged 70 and over have been established by the international Global Leadership Initiative on Malnutrition (GLIM) [9] in addition to those already existing for sarcopenia [10]. Reduced food intake is now one of the top five criteria to diagnose malnutrition [9], together with BMI, weight loss, muscle mass and inflammatory conditions. To assess food intake, the GLIM advocated use of semi-quantitative methods [9], as occurred in the NutritionDay^®^ survey [8].

Indeed, methods such as food diary, 24 h recall, food frequency questionnaires and Mini Nutritional Assessment (MNA^®^) [5,11] have shown limitations, especially in patients living in nursing homes, because of cognitive disorders or daily staff organization where time is often limited [12,13]. If healthcare providers could use easy and quick methods for nutritional assessment, this could certainly be a valuable contribution in organization and implementation of few but meaningful nutritional monitoring methods in inpatient settings [13,14,15]. Simple Evaluation of Food Intake (SEFI)^®^, combining a visual analogue scale [13,14] or evaluation of portions consumed [15], has established itself as a fast and reliable semi-quantitative method for diagnosis of nutritional risk or malnutrition. A single recent study [15] has shown that evaluation by medical staff of consumed food portions is feasible and reliable to screen malnutrition in a nursing home: if the patient ate more than 50% of their food portion, diagnosis of malnutrition could be ruled out [15]. To overcome cognitive difficulties of nursing home residents and lack of time for healthcare providers, photography associated with a posteriori evaluation of consumed food portions has also been developed [16]. A study in nursing homes showed that this method of photographing plates and/or trays before and after meals and performing a semi-quantitative assessment of food intake enabled reliable estimation of food intake whatever the type of food analyzed [17].

However, no study has yet assessed whether food intake of older people living in nursing homes was stable over time. Moreover, in these patients, the performance of the SEFI^®^ to diagnose malnutrition according to the GLIM criteria has never been studied. Therefore, our primary aim was to assess maintenance over one month of one-day semi-quantitative assessment of food intake, and our secondary aim was to assess its performance in diagnosing malnutrition in the older people living in a nursing home.

## 2. Materials and Methods

### 2.1. Study Design

This is a preliminary, prospective, monocentric and observational study. The patients were selected during the month of January 2022 within the nursing home Hôtel Dieu at Rennes University Hospital (CHU Rennes). This nursing home has a capacity of 120 beds divided in four units and welcomes a large number of individuals with Alzheimer’s disease or related dementia pathologies. One unit presented an incompatible organization for food monitoring: the meals were served dish by dish and not all on a tray. Thus, the residents were included in three units. Each of the three units was analyzed over a 4-week period, which includes an entire menu cycle of three weeks. Food intake and nutritional status were analyzed for each patient by one investigator (AB). The total duration of the study was four months.

### 2.2. Patient Selection

The patients included had to be aged 70 and over, have a limitation of independence affecting at least one activity of daily living (as measured by the Katz ADL scale [18]), be on passive or active diet, permanently reside in the nursing home and be affiliated to the national social-welfare system. The criteria for non-inclusion were a life expectancy estimated shorter than the duration of the study, presence of severe behavioral symptoms incompatible with performance of the nutritional assessment, prescription of exclusive enteral nutrition and an opposition to the study by the resident or a legal representative.

### 2.3. Assessment of Food Intake by the SEFI^®^ Assisted by Photography (SEFI^®^-AP)

In the nursing home, the meals provided are individualized to patient individual needs by a dedicated dietitian. The menu cycles are for 3 weeks. The food intake of lunch and dinner of each subject was followed up over four weeks, five days a week (Monday to Friday). Breakfast was not included for two reasons: (i) this is usually the most consumed meal of the day so not discriminative to identify patients at risk for malnutrition and worse outcomes, contrary to lunch or dinner [8]; (ii) because of logistical reasons, it was not possible to have an investigator monitor food intake for three meals. Therefore, we decided to focus on lunch and dinner.

One investigator (AB) photographed with a standard cell phone camera each patient’s meal trays before and after lunch and dinner. The camera was placed parallel to the tray and at a distance of about 40 cm. During 20 days, 2880 meals, i.e., 40 meals per patient, were photographed and analyzed. The consumed food portions were measured by the SEFI^®^ at the end of the follow-up by one single investigator (AB) because a previous study demonstrated good inter-observer reproducibility of SEFI^®^ and SEFI-AP^®^ [19]. The SEFI^®^ (Knoë-Groupe Get, Le Kremlin Bicêtre, France; www.sefi-nutrition.com, accessed on 26 January 2023) is a score rated from 0 (nothing is consumed) to 10 (everything is consumed) in whole numbers (Figure 1). A SEFI^®^ < 7/10 corresponds to decreased food intake and is associated with risk of malnutrition [14]. A SEFI^®^ ≥ 7 corresponds to satisfactory food intake. Day 3 was chosen as the reference day to allow a two-day period of adaptation to patients and healthcare providers regarding the presence of the observer investigator for monitoring food intake.

### 2.4. Assessment of Nutritional Status

Nutritional status was assessed in each patient during the first week of food intake monitoring according to the GLIM criteria. To calculate weight loss, weights measured one and six months ago were collected. Current weight was measured in clothed subjects using a patient lift or weigh chair. Height was collected from the medical record or was extrapolated from heel–knee distance to the nearest 1 cm, knee forming an angle of 90° with the thigh, using the Chumlea formula [20]. BMI was calculated by dividing mass in kilograms by height in meters squared. Calf circumference was measured using a measuring tape on both calves, knee at 90°, at the level of the largest calf diameter without compression soft tissues. Handgrip test was performed elbow at 90° placed on the armrest or the bed, three times on each arm, to assess muscle strength in patients able to do so. Only the highest value was retained. The presence of edema in the lower limbs was also noted. Body composition was assessed using bioimpedance analysis (BIA) at 50 kHz and a low intensity alternating current of 70 µA (Z-Metrix^®^ multifrequency impedancemeter, Bioparhom, Le Bourget du Lac, France) by one single investigator (AB). BIA was performed on subjects in dorsal decubitus, arms and legs apart on both sides of the chest, without contact between the trunk and the limbs, according to manufacturer instructions. The 50 kHz resistance, reactance and phase angle were measured. Fat mass, fat-free mass ((FFM), appendicular skeletal muscle mass (ASMM) according to the Sergi [21] equation and skeletal muscle mass (SMM) according to the Wang [22] and Janssen [23] equations (Table 1). The respective indexes were calculated dividing the FFM, ASMM or SMM values by height in meters squared (Table 1). The last available serum albumin was collected.

Malnutrition diagnosis was made according to the GLIM criteria for people over 70 years of age [9] based on identification of at least one phenotypic and one etiologic criterion. The etiologic criterion was the presence of at least one chronic disease in all the patients. The phenotypic criteria were at least one among the following: BMI < 22 and/or weight loss (≥5% in 1 month or ≥10% in 6 months) and/or presence of reduced muscle mass [21] using one of the methods presented in Table 1. Sarcopenia was defined by the association between decreased muscle function (handgrip grip strength < 16 kg for women and <27 kg for men, or inability to perform the test) and reduced muscle mass (Table 1) [10].

### 2.5. Other Data Collection

The clinical and demographic characteristics of the patients were collected from medical records: gender, age, main diagnosis, diseases that may impact the patient’s nutritional status (dementia, diabetes, organ failure, depression, cardiovascular disease, cancer, systemic disease), level of patient mobility (bedridden, semi-bedridden (including patients who can move independently in a wheelchair) or mobile) and swallowing disorders. The following information regarding treatments was collected: number of pills per day, number and nature of vitamins and/or minerals, drift of glucose and/or physiological saline. Additional information regarding diet was collected: high-protein and high-calorie diet and oral nutritional supplements.

### 2.6. Study Endpoints

The primary endpoint was daily food consumption evaluated by the SEFI^®^-AP over a day (lunch and dinner) or one single meal (lunch). The secondary endpoint was diagnosis of malnutrition according to the GLIM criteria.

### 2.7. Ethical Considerations

The study obtained the agreement of the CHU Rennes Clinical Ethics Committee No. 22.28 as a non-interventional study. Meal photography does not require any contact with the patients, and assessment of nutritional status is part of the recommended routine care in France. All the data collected were anonymized. An information notice was distributed allowing non-opposition of residents, their relatives or legal representatives. The study results will be communicated to the nursing home team and could lead to modifications in the professional practices around food.

### 2.8. Statistical Analyzes

Quantitative variables are expressed as mean ± standard deviation (SD) or median ± interquartile according to the value distribution. Data were compared using parametric Student’s *t*-test or non-parametric Mann–Whitney–Wilcoxon test. Qualitative variables are presented as number (n) and percentage (%). The groups were compared by parametric Chi2 (χ2) or non-parametric Fisher test. McNemar test or exact McNemar test if one of the modalities has a small number compared the data paired with two modalities. To study the maintenance of SEFI^®^-AP over time, we compared the SEFI^®^-AP measured on the third day (D3) with the SEFI^®^ medians of the day intervals considered. A generalized linear model on ranks with repeated measures evaluated the evolution of SEFI^®^-AP. Receiver operating characteristic (ROC) curves were used to analyze the association of SEFI^®^-AP with malnutrition diagnosis and to find the best cut-off associated with malnutrition diagnosis according to the Youden method. The cut-off was chosen by maximizing sensitivity and specificity using the Youden index (Y = Sensitivity + Specificity − 1). ROC curves were analyzed with their area under the curve (AUC) and 95% confidence interval (95% CI). Discriminative power of AUC was determined according to the following classification: 0.90 ≤ AUC ≤ 1.0, excellent; 0.80 ≤ AUC < 0.90, good; 0.70 ≤ AUC < 0.80, fair; 0.60 ≤ AUC < 0.70, poor; 0.50 ≤ AUC < 0.60, fail. Sensitivity, specificity, positive (PPV) and negative (NPV) predictive values as well as 95% CI were calculated. Spearman test was used for the correlation analyses. All statistical tests have a significance level of 0.05. Statistical analyses were performed using SAS software, v.9.4^®^ (SAS Institute, Cary, NC, USA).

## 3. Results

### 3.1. Patient Recruitment

The study flow chart is reported in Figure 2. Among the 90 subjects, 75 were eligible according to the non-inclusion and inclusion criteria. As three died before the start of the study, 72 patients were included (Figure 2). Food intake and nutritional status were analyzed in 70 and 69 patients, respectively. Lowered muscle strength was attributed to the 39 patients unable to perform the test and with reduced mobility.

### 3.2. Characteristics of the Study Population

The demographic, clinical and nutritional characteristics of the 70 included patients are shown in Table 2. The study population consisted of 54 women and 16 men. The individuals were between 72 and 100 years old, with an average age of 85.1 ± 6.4 years. Food intake was decreased, i.e., SEFI^®^-AP < 7, in 39% (n = 27/73) of patients: 22 women (40.7% of women) and five men (31.2% of men) had SEFI^®^ < 7 on day 3. The mean ± SD SEFI^®^-AP was 6.9 ± 2.8. Thirty-seven percent of patients needed help for meals. The main comorbidities were dementia (87%) and cardiovascular diseases (76%). Twenty-three percent of patients were obese (Table 2), and 36% were malnourished according to BMI. All but eight patients (11%) had serum albumin < 35 g/L. Patients with a SEFI^®^ < 7 had significantly lower calf circumference, lower BMI, lower fat-free mass index (but not fat mass index) and lower skeletal muscle mass, according to Sergi [22], Wang [23] or Janssen [24] equations (Table 2).

### 3.3. Maintenance over One Month of One-Day Semi-Quantitative Assessment of Food Intake

We studied the evolution over time of the assessment of food intake by the SEFI^®^-AP during the month following the first day of measurement using a generalized linear statistical model on ranks with repeated measures. The SEFI^®^-AP did not vary significantly from the first (day 1) to the twentieth day (day 20) of food intake monitoring, whether for lunch SEFI^®^-AP median (*p* = 0.12) or for the means of lunch and dinner SEFI^®^-AP median (*p* = 0.15) (Figure 3). Gender, level of mobility, help for meals, oral nutritional supplements, dementia, depressive syndrome and obesity were not associated with different evolution over time of the SEFI^®^-AP (data not shown). The lunch SEFI^®^-AP median (Figure 3A) and the means of lunch and dinner SEFI^®^-AP median (Figure 3B) turned around the 7/10 threshold. The median values of lunch SEFI^®^-AP or lunch and dinner SEFI^®^-AP means were relatively stable regardless of the considered food-monitoring period (Table 3). Given the standard deviations, the median SEFI^®^-AP of the means of lunch and dinner seemed more stable than the median SEFI^®^ of lunch alone (Table 3). At day 3, the median SEFI^®^-AP of lunch (*p* = 0.12) and the median SEFI^®^-AP of the means of lunch and dinner were very well correlated (Spearman rho coefficient = 0.83, *p* < 0.001).

### 3.4. Performance of Day 3 SEFI^®^-AP to Identify Decreased Food Intake during the Following Month

We, therefore, studied to what extent diagnosis of decreased food intake defined by the SEFI^®^-AP < 7 at day 3 was reproducible on the following 5, 10, 15 and 20 days. Day 3 SEFI^®^-AP < 7 or ≥ 7 was considered to be a good marker if SEFI^®^-AP < 7 or ≥7 was maintained for at least 70% of the time during the corresponding days. The sensitivity of day 3 SEFI^®^-AP to anticipate the food intake assessment during different periods was better for lunch and dinner than for lunch alone (Table 4). This was the contrary for specificity (Table 4). Sensitivity of day 3 SEFI^®^-AP was better to measure food intake during days 4 to 10 (89%) but remains correct during days 4 to 20 (78%) (Table 4). Specificity was best for days 4 to 15 and days 4 to 20 but was low (28–42%) (Table 4). The negative predictive value of day 3 SEFI^®^-AP to measure food intake was best during days 4 to 10 (79%). The positive predictive value was low (around 40%) for each studied period.

### 3.5. Malnutrition Prevalence According to Different GLIM Criteria

The proportion of malnourished patients varied according to the GLIM phenotypic criteria used (Table 5): BMI or weight loss detected 1.6% to 23.2% of malnourished individuals. Including reduced muscle mass as a phenotypic criterion, the proportion of malnourished individuals increased sharply to 36.2%, 46.4% and 47.8% using Sergi ASMM index, FFM index or calf circumference, respectively. One-hundred percent (19/19) of men and 98.1% (52/54) of women have reduced muscle strength. Sarcopenia was diagnosed in 37.3%, 23.9%, 20.9%, 6.0% and 34.8% using FFM index, Sergi ASMM index, Wang or Janssen SMM indexes or calf circumference to identify reduced muscle mass, respectively. Using calf circumference, Wang SMM index or FFM index, the proportion of patients with sarcopenia was significantly higher in the group of patients with day 3 SEFI^®^ < 7 than in the one with day 3 SEFI^®^ ≥ 7: 14 (51.9%) vs. 10 (23.8%), *p* = 0.02; 9 (34.6%) vs. 5 (12.2%), *p* = 0.03; 14 (53.8%) vs. 11 (26.8%), *p* = 0.03, respectively.

### 3.6. Performance of Day 3 SEFI^®^-AP for Diagnosis of Malnutrition

The performance of day 3 SEFI^®^-AP (mean of lunch and dinner) for diagnosis of malnutrition using calf circumference <31 cm as a phenotypic criterion was correct: AUC = 0.71 [95% CI, 0.59–0.83] (Figure 4). Day 3 SEFI^®^-AP sensitivity was better if ≤9.5 than <7, and inversely for specificity. For SEFI^®^-AP ≤ 9.5, sensitivity was 90.6% [95% CI, 80.5–100] (n = 29/32), specificity was 40.5% [24.7–56.4] (n = 15/37), positive predictive value was 56.9% [43.3–70.5] (n = 29/51) and negative predictive value was 83.3% [66.1–100] (n = 15/18). For SEFI^®^-AP < 7, sensitivity was much lower (56.3% [39.1–73.4]) (n = 18/32), specificity was greater (64.9% [49.5–80.2]) (n = 24/37), the positive predictive value was 58.1% [40.7–75.4] (n = 18/31) and the negative predictive value was 63.2% [47.8–78.5] (n = 24/38).

The performance of day 3 SEFI^®^-AP at lunch alone was lower (AUC = 0.64 [0.50–0.78]). The performance of day 3 SEFI^®^-AP (mean of lunch and dinner) for diagnosis of malnutrition was poor if the phenotypic criteria chosen were FFM index (AUC= 0.62 [0.49–0.75]) or Sergi ASMM index (AUC = 0.64 [0.50–0.78]).

## 4. Discussion

Our study, conducted in older people living in a nursing home, showed good stability of food intake over time. One-day assessment of food intake by the SEFI^®^-AP correctly reflects food intake during the following month and diagnosis of malnutrition. The SEFI^®^-AP sensitivity for malnutrition diagnosis was better if ≤9.5 than <7. These results suggest that any decrease in food intake should lead to suspect malnutrition.

In older people living in nursing homes, one-day assessment of food intake by the SEFI^®^-AP is stable during the month following measurement. This finding is new. The performance of the one-day SEFI^®^ to identify decreased food intake during the following month is correct and better if considering lunch and dinner rather than lunch alone. This means that some patients are possibly able to compensate for a low food intake during a meal by a higher food intake at the next meal, and inversely. It is important to note that the comparisons of day 3 SEFI^®^-AP with food intake monitoring of one, two or three weeks (i.e., days 4 to 10, 4 to 15 and 4 to 20) revealed only very few differences in the values of sensitivity, specificity, positive and negative predictive values. This indicates that a 3-week food intake evaluation is not at risk of being different from that carried out over one week. Therefore, for daily practice in nursing homes, we could advise to repeat monitoring of food intake of lunch and dinner only one day every three weeks. However, the workload of nursing home healthcare providers may be a brake for performing this sequential monitoring of food intake, even every three weeks, and even based on photography. Some authors have proposed to use artificial intelligence to overcome these limitations. Software for analyzing food intake automatically from photography of meal trays before and after a meal has been developed. This requires only little human intervention [26]. This method provides each patient an automated energy intake calculation and long-term monitoring, including an alert system in the event of decreased food intake. Preliminary results are promising, but the results need to be confirmed on a large scale [26]. With this perspective, our study provides important information on timing for dietary monitoring.

Our study indicates that one-day SEFI^®^-AP could detect patients remaining with decreased food intake in the days following assessment. The better sensitivity (up to 80–90%) than specificity means that SEFI^®^-AP has correct performance to detect at day 3 patients who will truly maintain decreased food intake during the following days. The better negative (up to 70–80%) than positive predictive values mean that the proportion of patients maintaining sufficient food intake during the following days if the SEFI^®^-AP at day 3 is ≥7 is higher than the proportion of patients having decreased food intake during the following days if the SEFI^®^-AP at day 3 is <7. The low specificity suggests that a SEFI^®^-AP at day 3 ≥ 7 fails to identify correctly the patients who will have satisfactory food intake over time. This also means that there is a high proportion of subjects with false positive SEFI^®^-AP at day 3. However, from a practical point of view, this would lead to monitoring closely patients who are at apparent risk of malnutrition and those who are finally not. It is less prejudicial than believing food intake is normal when it is not, thus exposing the patient to malnutrition.

An important finding of our study is that one-day SEFI^®^-AP expressed as the mean of lunch and dinner appropriately diagnosed malnutrition if the chosen GLIM phenotypic criterion was calf circumference <31 cm. The SEFI^®^-AP sensitivity was better if ≤9.5 (91%) than <7 (56%), and inversely for specificity (40% and 65%, respectively). This raises the question of the reliability of the threshold of 7 for diagnosis of malnutrition in older people living in a nursing home. Previous studies have shown that a SEFI^®^ threshold of 7 was reliable to detect risk of malnutrition in hospitalized and ambulatory patients under the age of 75 [13] or in primary care [14]. In a previous study performed in a nursing home, we found that, in people eating more than 50% of their served food, diagnosis of malnutrition based on the MNA^®^ could be ruled out [15]. In the present study, a SEFI^®^-AP < 7 does not perform well for malnutrition diagnosis, whereas ≤9.5 does. Good sensitivity of SEFI^®^-AP ≤9.5 suggests that any decrease in food intake should lead to suspect malnutrition in older people living in nursing homes. This does not seem surprising if we consider the food catering practices in nursing homes: indeed, to avoid food wastage, patients who are ‘small eaters’ receive smaller food portions, whereas ‘normal eaters’ receive normal portions. For example, a small eater eating 80% of the completely served portion could in fact experience decreased food intake, i.e., energy intake < nutritional needs, and be at malnutrition risk, whereas the SEFI^®^ is above 7/10.

Nevertheless, our study confirmed the findings [15] that evaluation of consumed food portions by the SEFI^®^ is a reliable method for malnutrition diagnosis in geriatric institutions. The European Society for Clinical and Metabolism notably recommends to monitor food intake to identify risk of malnutrition in older people [5] and hospitalized patients [27]. Our study confirms that the SEFI^®^ could be used for that purpose. The photography assistance, i.e., SEFI^®^-AP, should help to overcome the limitations related to nursing home organization: large number of observers, time constraints, workload, evaluation biases caused by subjectivity and inability of patients with neurocognitive disorders to evaluate their food intake themselves [17]. In any case, malnutrition diagnosis needed to be confirmed using the GLIM criteria [5,9]. It is now necessary to test in a clinical trial whether dedicated nutritional management in older patients living in nursing homes with SEFI^®^-AP ≤9.5 could improve their food intake and avoid malnutrition or its worsening.

Our study confirms previous data from the literature that have already demonstrated the correlation between reduced food intake and malnutrition, itself related to increased morbidity and mortality [8,12,13]. Low BMI, low calf circumference, reduced muscle mass measured by BIA and the presence of sarcopenia were significantly associated with a SEFI^®^-AP < 7. However, the SEFI^®^-AP performance for malnutrition diagnosis is correct only if the phenotypic criterion is calf circumference. It is poor if it is BIA-measured muscle mass. It remains difficult to draw firm conclusions as the AUC confidence intervals are very wide and the studied population is small. However, the inherent characteristics of older people living in nursing homes (e.g., advanced age, chronic diseases, dementia, multiple comorbidities) could explain this result. Indeed, all the characteristics could themselves be causes of malnutrition independently of any reduction in food intake. Our study was not designed to study whether decreased food intake was associated with higher morbidity and mortality, but this could be completed in the future to confirm previous data in geriatric institutions [8]. By conducting the same methodology of one-day food monitoring, our study results are in line with those of the NutritionDay^®^ survey, indicating that decreased food intake with a single meal at hospital is associated with higher mortality one month later [8].

Our study suggests that assessment of muscle strength by handgrip appears to be of little relevance in older people living in nursing homes. Given the high level of dependency of the patients, all patients have decreased muscle strength regardless of their nutritional status or muscle mass. The GLIM group has recently suggested to exclude muscle strength from the malnutrition diagnosis criteria but advises to assess it systematically independently of the nutritional assessment [21]. The proportion of patients with sarcopenia we found varied between 6% and 35%, in accordance with the findings of a published meta-analysis [6].

Our study provides new insights on the GLIM phenotypic criteria to use for malnutrition diagnosis in older people living in nursing homes. We found that BMI or weight loss alone detected only a small proportion of malnourished older individuals. Yet, the NutritionDay^®^ survey demonstrated the importance of BMI in follow-up of patients living in nursing homes [2]. This can be explained in particular by the fact that aging is characterized by a reduction in fat-free mass without necessarily a reduction in BMI and weight [25]. In addition, the nursing home population is characterized by stability in terms of state of health, and, therefore, possibly very gradual and inconspicuous weight loss. This, therefore, confirms the importance of combining different parameters to detect malnutrition in people aged 70 and over [9]. To this aim, our study suggests that measurement of muscle mass by BIA does not perform better than calf circumference measurement to diagnose malnutrition or sarcopenia. This result seems particularly relevant in the setting of a nursing home, where calf circumference is easier to implement than BIA.

Our study has strengths and limitations. The main strength is the duration and exhaustiveness of the food intake monitoring, allowing photographing and assessing 2′880 meals. There are few missing data. The nutritional assessment was based on the GLIM recommendations, sarcopenia diagnosis and an exhaustive assessment of muscle mass, including several validated equations and calf circumference. Diseases, comorbidities and type of diet were not considered as non-inclusion criteria. One limitation is that breakfast was not included, but this may have not biased the results [8]. One other limitation is the monocentric design, raising the question of representativeness of the population studied and generalization of the results observed. Indeed, there is great variability in malnutrition prevalence between nursing homes, from 1.5% to 66.5% [1]. In this study, the proportion of malnourished individuals is around 40%. In addition, the studied nursing home is public and housed a higher proportion of patients with severe cognitive disorders than other nursing homes (80% vs. 48% [28]). However, these diseases have a real impact on food intake [29], which makes us consider that this nursing home was a good choice regarding our study aims. Only one patient with severe psychotic disorder was excluded, so this may not have biased the results. In addition, the size of the study population was small, so the study lacked power to assess the diagnostic performance of SEFI^®^ for malnutrition, but this was not the main objective of the study. Our study lacks an evaluation of food appreciation as the unappetizing character of food [19] could have influenced food intake independently of nutritional status. Another limitation concerns the reliability of the BIA equations and thresholds used [21,22,23,24], which have been validated in healthy adults or older people living independently at home. Thus, their extrapolation to patients who are all highly dependent, mostly bedridden, with extremely limited physical activity and ill may be underlined. Nevertheless, there are currently no validated BIA equations for nursing home residents. Evaluation of stability of SEFI^®^ over time was carried out from the medians of the SEFI^®^ of all the patients, which can, therefore, attenuate possible intraindividual variability in food intake. We remedied this by measuring the performance of day 3 SEFI^®^-AP for evaluating maintenance of the initial level of food intake in the following month. Finally, to apply the photographic method used, all the meal dishes had to be served and cleared at the same time on a tray. However, food service is not always organized this way in nursing homes. Dishes can be served and cleared one by one, which prevents taking a photograph of the entire meal tray. This practice notably led to exclusion of one nursing home unit from the study. This limitation suggests that applicability in current practice of this assessment method of food intake may vary according to the food service organization.

## 5. Conclusions

Our study reports that food intake of older people living in nursing homes is stable over one month. One-day SEFI^®^-AP correctly anticipates food intake during the following month and correctly predicts diagnosis of malnutrition. Any decrease in food intake should lead to suspect malnutrition. Monitoring of food intake based on one-day meal photography and SEFI^®^ would help to improve identification of residents who are malnourished or at risk of malnutrition regardless of their underlying pathologies and, thus, trigger earlier the nutritional care to fight against malnutrition. These study findings open up promising avenues for simplifying and improving methods of food intake monitoring and screening for malnutrition in older people living in nursing homes.

## Figures and Tables

**Figure 1 nutrients-15-00646-f001:**
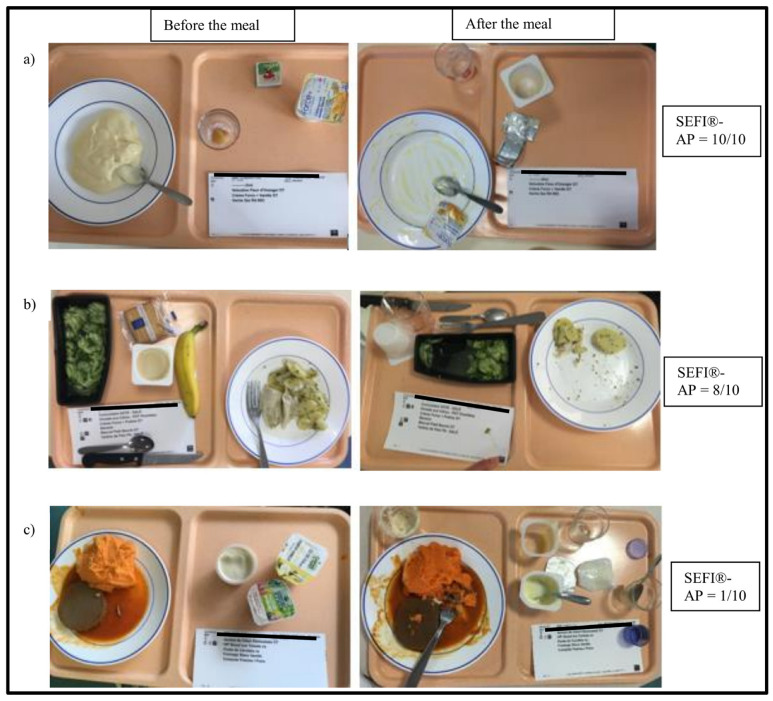
Three examples (**a**–**c**) of scoring of the Simple Evaluation of Food Intake^®^ assisted by photography (SEFI^®^-AP). Photographs of the meal trays before (left column) and after food intake (right column) were analyzed and the SEFI^®^-AP was scored by comparing the served to consumed food portions. A/Meal tray completely consumed: SEFI^®^-AP = 10/10. B/More than ¾ of the meal tray consumed: SEFI^®^-AP = 8/10. C/Less than ¼ of the meal tray consumed: SEFI^®^-AP = 1/10.

**Figure 2 nutrients-15-00646-f002:**
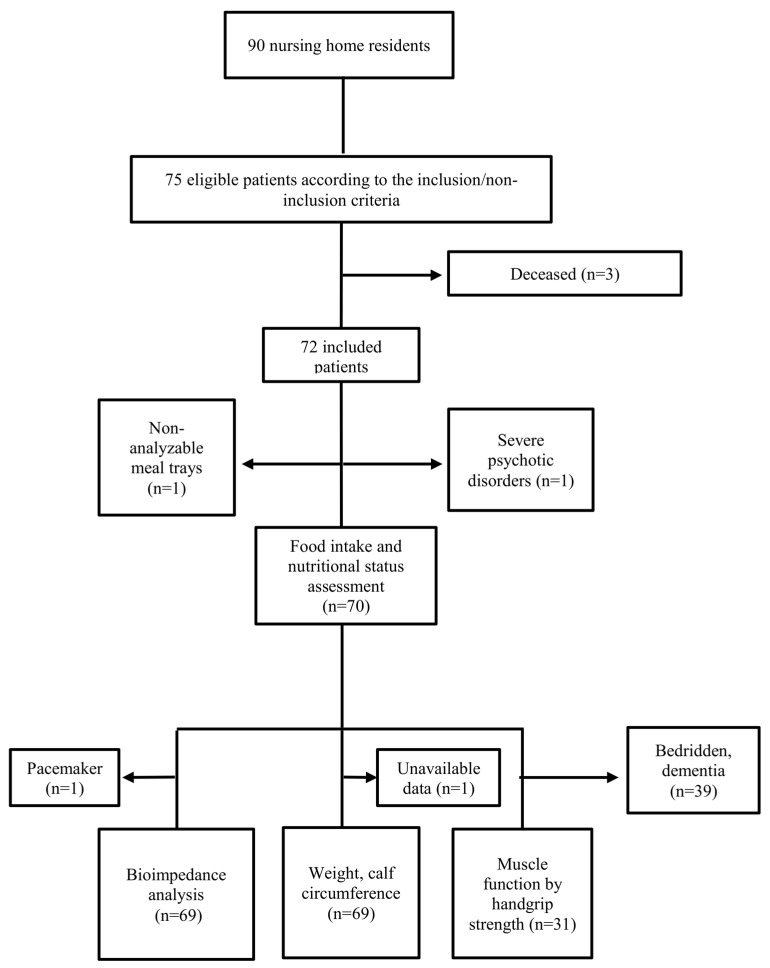
Study flow chart. n, number of patients.

**Figure 3 nutrients-15-00646-f003:**
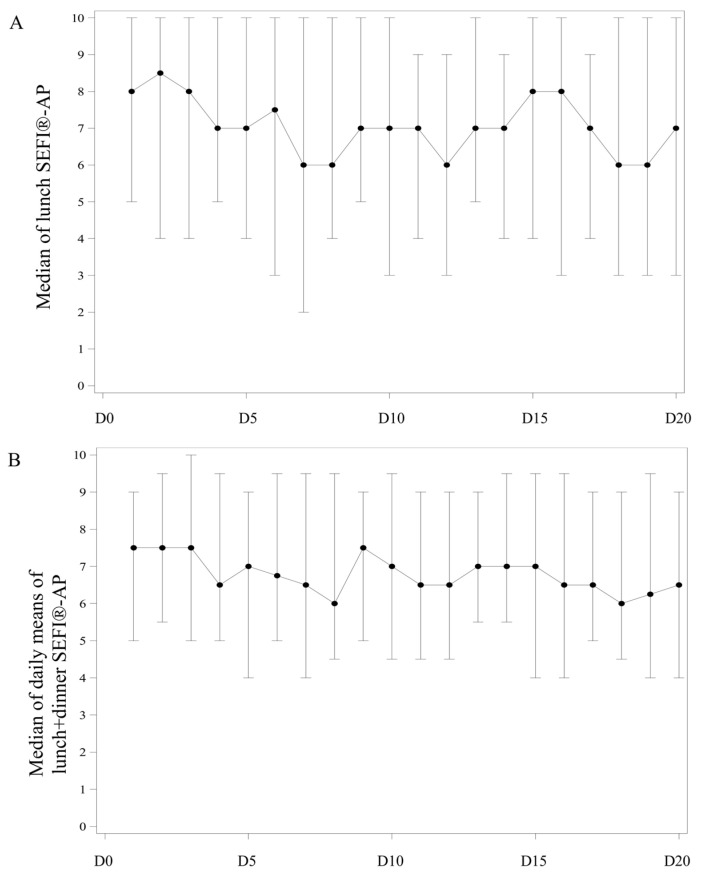
Evolution of the Simple Evaluation of Food Intake^®^ assisted by photography (SEFI^®^-AP) between day (D) 0 and D 20 (n = 70). (**A**) Median SEFI^®^ of lunch. (**B**) Median SEFI^®^-AP of the means of daily lunch and dinner. Each bar represents median and interquartile of the SEFI^®^-AP score ranging from 0 to 10 each day from D0 to D20.

**Figure 4 nutrients-15-00646-f004:**
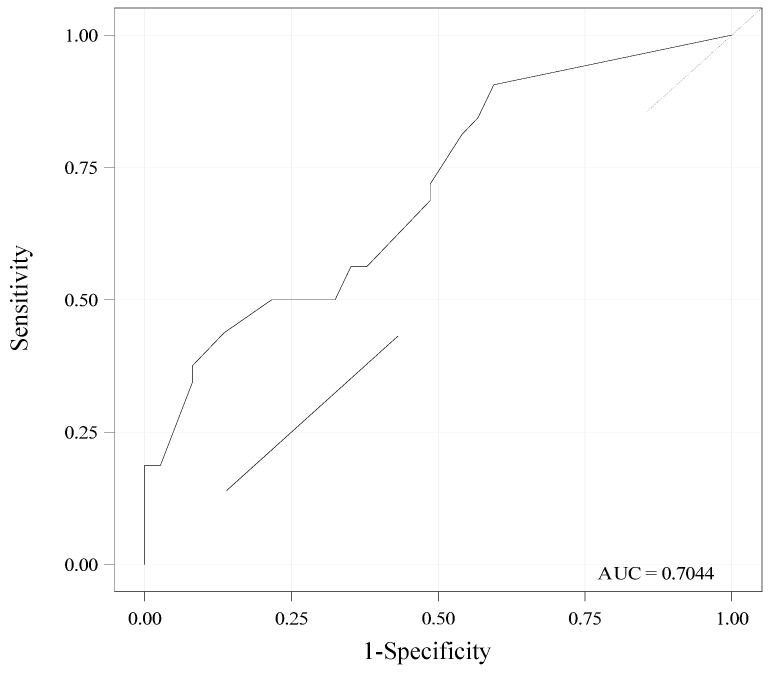
Area under the receiving operating curve (ROC) showing the performance of the Simple Evaluation of Food Intake^®^ assisted by photography (SEFI^®^-AP) at day 3 for diagnosis of malnutrition (n = 69). SEFI^®^-AP is expressed as the mean of lunch and dinner. Malnutrition was diagnosed according to the Global Leadership Initiative on Malnutrition (GLIM) criteria: body mass index < 22 or weight loss at 1 month ≥ 5% or weight loss at 6 months ≥ 10% or lower calf circumference (<31 cm). AUC = 0.70 [95% confidence interval, (CI) 0.58–0.83] indicates a correct diagnostic performance. Sensitivity, specificity, positive and negative predictive values [95% CI] of the SEFI^®^-AP < 7 at day 3 for diagnosis of malnutrition were: 56% [39–73], 65% [49–80], 58% [41–75] and 63% [48–78], respectively.

**Table 1 nutrients-15-00646-t001:** Methods for muscle mass assessment and thresholds used for determining reduced muscle mass according to the Global Leadership Initiative on Malnutrition [9] recommendations.

Methods for Assessing Muscle Mass	Equations	Thresholds for Reduced Muscle Mass	
		Men	Women
Calf circumference (cm)	-	<31	<31
ASMM (kg) (Sergi equation) [22]	= 3.964 + (0.227 × RI) + (0.095 × weight(kg)) + (1.384 × gender) + (0.064 × Xc)	<20	<15
ASMM index (kg/m²) (Sergi equation) [22]	= ASMM of Sergi (kg)/height (m)²	<7	<5.5
Skeletal Muscle Mass (SMM) (kg) (Wang equation) [22]	= (0.0093 × TBK) − (1.31 × gender) + (0.59 × black) + (0.024 × age) − 3.21	<20	<15
SMM index (Wang equation) (kg/m²) [23]	= SMM Wang(kg)/height (m) ²	<7	<5.7
SMM (kg) (Janssen equation) [24]	= ((height(cm)²/R × 0.401) + (gender × 3.825) + (age × −0.071)) + 5.102	<20	<15
SMM index (kg/m²) (Janssen equation) [24]	= SMM of Janssen(kg)/height (m)²	<7	<5.7
FFM index (kg/m²) [25]	= FFM (kg)/height (m)²	<17	<15

Gender: 0 if woman, 1 if man/1 if African American, 0 otherwise. ASMM, appendicular skeletal muscle mass; FFM, fat-free mass; R, resistance; RI, resistance index; SMM, skeletal muscle mass; TBK, total body potassium; Xc, reactance.

**Table 2 nutrients-15-00646-t002:** Clinical, demographic and nutritional characteristics of the included population (n = 70) according to the levels of food intake at day 3 measured by the Simple Evaluation of Food Intake^®^ assisted by photography (SEFI^®^-AP) (mean of lunch and dinner).

Variables	Total Population (n = 70)	day 3 SEFI^®^-AP <7 (n = 27)	≥7 (n = 43)	*p*
** *Demographics* **				
**Gender**				0.49
Women	54 (77.1%)	22 (81.5%)	32 (74.4%)	
Men	16 (22.9%)	5 (18.5%)	11 (25.6%)	
Age (years)	85.1 ± 6.4	86.4 ± 5.1	84.3 ± 7.0	0.19
**Comorbidities**				
Dementia	61 (87.1%)	22 (81.5%)	39 (90.7%)	0.29
Cancer	7 (10.0%)	5 (18.5%)	2 (4.7%)	0.1
Organ failure	18 (25.7%)	4 (14.8%)	14 (32.6%)	0.1
Cardiovascular disease	53 (75.7%)	20 (74.1%)	33 (76.7%)	0.8
Diabetes	9 (12.9%)	4 (14.8%)	5 (11.6%)	0.73
Depressive syndrome	11 (15.7%)	5 (18.5%)	6 (14.0%)	0.74
**Diet characteristics**				
Assistance for food intake	26 (37.1%)	9 (33.3%)	17 (39.5%)	0.60
HPHC diet	33 (47.1%)	15 (55.6%)	18 (41.9%)	0.26
ONS	33 (47.1%)	14 (51.9%)	19 (44.2%)	0.53
**Treatment**				
Micronutrients	62 (88.6%)	25 (92.6%)	37 (86.0%)	0.473
Glucose SC infusion	24 (34.3%)	7 (25.9%)	17 (39.5%)	0.24
NaCl SC infusion	23 (32.9%)	6 (22.2%)	17 (39.5%)	0.13
≥5 drugs	59 (84.3%)	23 (85.2%)	36 (83.7%)	1.00
**Nutritional evaluation**				
BMI(kg/m²)	26.1 ± 5.5	24.2 ± 5.0	27.4 ± 5.4	**0.02**
22 ≤ BMI < 25 (normal weight)	11 (15.9%)	2 (7.4%)	9 (21.4%)	0.18
25 ≤ BMI < 30 (overweight)	26 (37.7%)	14 (51.9%)	12 (28.6%)	**0.05**
BMI ≥ 30 (obesity)	16 (23.2%)	2 (7.4%)	14 (33.3%)	**0.01**
BMI < 22 (moderate malnutrition)	16 (23.2%)	9 (33.3%)	7 (16.7%)	0.11
BMI < 20 (severe malnutrition)	9 (13.0%)	8 (29.6%)	1 (2.4%)	**0.001**
Calf circumference (cm)	33.1 ± 5.0	31.1 ± 4.3	34.5 ± 5.1	**0.005**
** *Bioimpedance analysis* **				
FM (% of weight)	36.57 ± 9.53	35.73 ± 9.13	37.59 ± 9.91	0.3781
FM index (kg/m²)	9.82 ± 3.83	8.72 ± 3.25	10.68 ± 4.05	0.0630
FFM (% of weight)	63.43 ± 9.53	64.27 ± 9.13	62.41 ± 9.91	0.3781
FFM index (kg/m²)	16.26 ± 2.97	14.96 ± 2.36	16.97 ± 3.06	**0.0013**
SMM index (Wang equation) (kg/m²)	6.79 ± 1.44	6.14 ± 0.95	7.13 ± 1.53	**0.0057**
ASMM index (Sergi equation) (kg/m²)	6.51 ± 1.08	5.96 ± 0.77	6.86 ± 1.10	**0.0005**
SMM index (Janssen equation) (kg/m²)	8.08 ± 1.60	7.39 ± 1.11	8.51 ± 1.72	**0.0063**
Total body water (% of weight)	45.58 ± 6.15	46.16 ± 5.97	45.32 ± 6.51	0.6068
Phase angle (degree)	3.94 ± 0.61	3.81 ± 0.64	4.04 ± 0.58	0.2143

Decreased food intake was defined by day 3 SEFI^®^-AP < 7. The qualitative variables are expressed in numbers (%) and compared using the χ2 or Fisher tests. The quantitative variables are expressed as mean ± standard deviation and compared using Student’s *t*-test or, for the bioimpedance analysis data, the Mann–Whitney–Wilcoxon test. Comorbidities were chosen as they are associated with risk for malnutrition. ASMM, appendicular skeletal muscle mass; BMI, body mass index; FM, fat mass; FFM, fat-free mass; HPHC, high-protein high-calorie; NaCl, sodium chloride; ONS, oral nutritional supplements; SC, subcutaneous; SMM, skeletal muscle mass.

**Table 3 nutrients-15-00646-t003:** Median values of the Simple Evaluation of Food Intake^®^ assisted by photography (SEFI^®^-AP) evaluated from lunch or lunch + dinner according to the follow-up period.

Follow-Up Period	SEFI^®^-AP (n = 70) Lunch	Mean of Lunch and Dinner
D3	6.8 ± 3.5 (0; 4; 8; 10; 10)	6.9 ± 2.8 (0.5; 5; 7.5; 10; 10)
D1 to D5	6.8 ± 3.1 (1; 4.5; 8; 10; 10)	7 ± 2.4 (1.5; 5; 7; 9.5; 10)
D1 to D10	6.7 ± 3.0 (1; 5; 7; 9; 10)	6.9 ± 2.4 (1; 5; 7; 9.3; 10)
D1 to D15	6.5 ± 2.9 (1; 5; 7; 9; 10)	6.9 ± 2.4 (1; 5; 7; 9.5; 10)
D1 to D20	6.4 ± 2.8 (1; 5; 6; 9; 10)	6.8 ± 2.4 (1; 4.9; 6.6; 9.3; 10)

The SEFI^®^-AP is expressed as a whole number on a scale ranging from 0 (nothing was eaten) to 10 (everything was eaten). Data are expressed as median ± interquartile (min; Q1; median; Q3; max). ‘Lunch’ corresponds to the median SEFI^®^-AP of the daily lunches. ‘Mean of lunch and dinner’ corresponds to the median SEFI^®^-AP of the means of daily lunches and dinners. D, day.

**Table 4 nutrients-15-00646-t004:** Sensitivity, specificity, positive and negative predictive values of the Simple Evaluation of Food Intake^®^ assisted by photography (SEFI^®^-AP) at day 3 to anticipate food intake during the following 5, 10, 15 or 20 days (n = 70).

SEFI^®^-AP	TP	FP	FN	TN	Sensitivity [95% CI]	Specificity [95% CI]	PPV [95% CI]	NPV [95% CI]
D4-D10 (L)	21	27	6	16	78% [62–94]	37% [23–52]	44% [30–58]	73% [54–91]
D4-D10 (LD)	24	32	3	11	89% [77–100]	26% [13–39]	43% [30–56]	79% [57–100]
D4-D15 (L)	17	25	10	18	63% [45–81]	42% [27–57]	41% [26–55]	64% [47–82]
D4-D15 (LD)	20	30	7	13	74% [58–91]	30% [17–44]	40% [26–54]	65% [44–86]
D4-D20 (L)	17	25	10	18	63% [45–81]	42% [27–57]	40% [26–55]	64% [47–82]
D4-D20 (LD)	21	30	6	13	78% [62–94]	30% [17–44]	41% [28–55]	68% [48–89]

SEFI^®^-AP at day 3 <7 or ≥ 7 is considered to be a good marker if SEFI^®^ < 7 or ≥ 7 is maintained for at least 70% of time during the corresponding days. The data of false negative (FN), false positive (FP), true negative (TN) and true positive (TP) are expressed as number. CI, confidence interval; D, day; L, median of lunch; LD, medians of means of lunch and dinner; NPV, negative predictive value; PPV, positive predictive value.

**Table 5 nutrients-15-00646-t005:** Proportion of patients (n = 69) being malnourished according to the Global Leadership Initiative for Malnutrition phenotypic criteria considered, and the Simple Evaluation of Food Intake^®^ assisted by photography (SEFI^®^-AP) at day 3.

GLIM Malnutrition Criteria	Total Population (n = 69)	SEFI^®^-AP at Day 3 < 7 (n = 27)	≥7 (n = 42)	*p*
BMI < 22	16 (23.2%)	9 (33.3%)	7 (16.7%)	0.1094
BMI < 20	9 (13.0%)	8 (29.6%)	1 (2.4%)	**0.0017**
Weight loss at 1 month ≥ 5%	2 (3.2%)	0 (0.0%)	2 (5.3%)	0.5136
Weight loss at 1 month ≥ 10%	1 (1.6%)	0 (0.0%)	1 (2.6%)	1.0000
Weight loss at 6 months ≥ 10%	4 (6.1%)	1 (4.0%)	3 (7.3%)	1.0000
Weight loss at 6 months ≥ 15%	0	0	0	-
Phenotypic criterion of reduced muscle mass: Low FFMI	33 (47.8%)	15 (55.6%)	18 (42.9%)	0.3027
Or low ASMM index (Sergi equation)	25 (36.2%)	11 (40.7%)	14 (33.3%)	0.5321
Or low calf circumference	32 (46.4%)	16 (59.3%)	16 (38.1%)	0.0853

Decreased food intake was defined by day 3 SEFI^®^-AP < 7. Figures are numbers (%) and compared using the χ2 or Fisher tests. For the reduced muscle mass criterion, muscle mass was measured by bioimpedance analysis using the fat-free mass index (FFMI) (fat-free mass (kg)/height (m)/^2^), the appendicular skeletal muscle mass (ASMM) according to the Sergi equation [21]) or by calf circumference. Muscle mass was considered as reduced if FFMI < 17 kg/m^2^ in men and <15 kg/m^2^ in women or ASMM <7 kg/m^2^ in men and <5 kg/m^2^ in women or calf circumference <31 cm in men and women. BMI, body mass index; FFM, fat-free mass; SMM skeletal muscle mass.

## Data Availability

Data are not publicly available due to ethical restrictions.

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
