# Peer review of "Assessment of Food Intake Assisted by Photography in Older People Living in a Nursing Home: Maintenance over Time and Performance for Diagnosis of Malnutrition"

_nutrients, 2023, doi:10.3390/nu15030646_

Round 1
Reviewer 1 Report
Thank you for your preliminary, prospective, monocentric and observational study showing that one-day nutritional monitoring at three-week intervals shows very little difference in sensitivity, specificity, positive predictive value and negative predictive value compared to a 3-week monitoring. This result can certainly be a valuable contribution in the organisation and implementation of few but equally meaningful nutritional monitorings in inpatient settings, where time is often limited. I only understood this importance in the discussion, perhaps this important motivation should be further elaborated in the introduction and the relevance should be supported with literature.
My main concern is that I cannot understand the calculation of the threshold. In addition, how does the area under the curve (AUC) determine the power? And please describe further the relation for the sensitivity and specificity!
Some minor comments:
Why it was day 3 that was used as the reference day against which the following average intake was compared? Perhaps more information could be given here.
The performance of day 3 SEFI®-AP at lunch alone was lower than at mean of lunch and dinner. I wonder why breakfast was not included, because, as you have justified, it is common to compensate for low consumption of one meal with other meals.
I am not quite sure why the nutritional status was collected 6 months before and 1 month before the nutritional assessment instead of after the nutritional assessment.
The reference in line 242 shows an error (Error! reference source not found).
Does the statement in lines 358-360 that there are better negative predictions than positive predictions contradict the next sentence that there are many false positive predictions? An initial definition of the terms would help me.
The SEFI®-AP diagnosis of malnutrition is only considered reliable if the phenotypic criterion is the calf circumference. Here I did not understand why the BIA measurement should be less reliable as a reverse conclusion, only because the results of the SEFI®-AP would be different when using the BIA measurement as phenotypic criterion.
Please discuss if prediction is possible if you identified that consume is stable in the setting.
Information on informed consent is strange. You argue that no personal contact required informed consent but what about conducting the Katz, the body weight and height etc for the GLIM, or the calf circumference and hand grip strength? Albumin level? This is part of the routine care? Please provide further details of this nursing home doing this kind of comprehensive assessment! This is unbelievable!

Author Response
Comments from the Editors and Reviewers:
Reviewer #1:
Thank you for your preliminary, prospective, monocentric and observational study showing that one-day nutritional monitoring at three-week intervals shows very little difference in sensitivity, specificity, positive predictive value and negative predictive value compared to a 3-week monitoring. This result can certainly be a valuable contribution in the organisation and implementation of few but equally meaningful nutritional monitorings in inpatient settings, where time is often limited. I only understood this importance in the discussion, perhaps this important motivation should be further elaborated in the introduction and the relevance should be supported with literature.
Author’s response:
We thank the Reviewer for her/his kind appreciation of our work and comments that have helped us to improve our manuscript. The paragraph in lines 57-63 has been revised as follows:
“Indeed, methods such as food diary, 24-hour recall, food frequency questionnaires and Mini Nutritional Assessment (MNA®) [5,11] have shown limitations, especially in patients living in nursing homes, because of cognitive disorders or daily staff organization where time is often limited [12,13]. If healthcare givers could use easy and quick methods for nutritional assessment, this could certainly be a valuable contribution in the organization and implementation of few but equally meaningful nutritional monitoring in inpatient settings [13-15].”
My main concern is that I cannot understand the calculation of the threshold. In addition, how does the area under the curve (AUC) determine the power?
Author’s response:
The Youden method is a well-known method to determine cut-offs, and the AUC value is determining the discriminative power of the test. We have clarified the methodology in the ‘Statistical analyzes’ paragraph.
The original text “Receiver Operating Characteristic (ROC) curve evaluated the diagnostic performance of the SEFI®-AP to predict food intake the day intervals considered and malnutrition diagnosis. The threshold was chosen by maximizing sensitivity and specificity using the Youden index (Y = Sensitivity + Specificity – 1). The area under the curve (AUC) determines the power of the test. Sensitivity, specificity, positive (PPV) and negative (NPV) predictive values as well as 95% confidence interval (95% CI) were calculated.” has been revised as follows:
“Receiver Operating Characteristic (ROC) curves were used to analyse the association of the SEFI®-AP with the malnutrition diagnosis, and to find the best cut-off associated with the malnutrition diagnosis according to the Youden method. The cut-off was chosen by maximizing sensitivity and specificity using the Youden index (Y = Sensitivity + Specificity – 1). ROC curves were analysed with their area under the curve (AUC) and its 95% confidence interval (95% CI). Discriminative power of AUC was determined according to the following classification: 0.90≤AUC≤1.0, excellent; 0.80≤AUC<0.90, good; 0.70≤AUC<0.80, fair; 0.60≤AUC<0.70, poor; 0.50≤AUC<0.60, fail. Sensitivity, specificity, positive (PPV) and negative (NPV) predictive values as well as 95% CI were calculated.” (lines 198-207).
And please describe further the relation for the sensitivity and specificity!
Author’s response:
The discussion paragraph in lines 368-381 has been revised accordingly:
“Our study indicates that SEFI®-AP one given day could detect the patients remaining with decreased food intake in the days following the assessment. The better sensitivity (up to 80-90%) than specificity means that SEFI®-AP has correct performance to detect at day 3 the patients who will truly keep decreased food intake during the following days. The better negative (up to 70-80%) than positive predictive values means that the proportion of patients keeping a sufficient food intake during the following days if the SEFI®-AP at day 3 ≥7 is higher than the proportion of patients having decreased food intake during the following days if the SEFI®-AP at day 3 <7. The low specificity suggests that a SEFI®-AP at day 3 ≥7 fails to identify correctly the patients who will have a satisfactory food intake over time. This also means that there is a high proportion of subjects with false positive SEFI®-AP at day 3. However, from the practical point of view, this would lead to monitor closely patients who are supposed at risk of malnutrition, i.e. with SEFI®-AP<7, and who are finally not. It is less prejudicial than believing food intake is normal whereas it is not, thus exposing the patient to malnutrition.”
Some minor comments:
Why it was day 3 that was used as the reference day against which the following average intake was compared? Perhaps more information could be given here.
Author’s response:
We have chosen to use day 3 as the reference day to let a two-day period of adaptation to patients and healthcare givers regarding the presence of the observer investigator for monitoring the food intake. We have added this information in the ’methods’ section (lines 121-123).
The performance of day 3 SEFI®-AP at lunch alone was lower than at mean of lunch and dinner. I wonder why breakfast was not included, because, as you have justified, it is common to compensate for low consumption of one meal with other meals.
Author’s response:
We have chosen to not include breakfast for two reasons:
- this is usually the most consumed meal of the day so not discriminative to identify patients at risk for malnutrition and worse outcomes contrary to lunch or dinner (Hiesmayr et al, ref [8])
- because of logistical reasons, it was not possible to have an investigator to monitor food intake for three meals. So based on [8], we decided to focus on lunch and dinner.
We have added this information in the ’methods’ section (lines 106-110). As also recommended by the Reviewer 3, the ‘study limitations’ section has been revised accordingly: “One limitation is that breakfast was not included, but this may have not biased the results [8] (lines 457-458).”
I am not quite sure why the nutritional status was collected 6 months before and 1 month before the nutritional assessment instead of after the nutritional assessment.
Author’s response:
We apologize for the writing. We would like meaning that we have collected in the patient medical record her/his weight measured one month and six months ago, to calculate weight loss that is part of the nutritional assessment. We have added this information in the ’methods’ section (lines 132-133): “To calculate weight loss, the weight measured one and six months ago were collected.”
The reference in line 242 shows an error (Error! reference source not found).
Author’s response:
This has been corrected (line 255) into ‘Table 3’.
Does the statement in lines 358-360 that there are better negative predictions than positive predictions contradict the next sentence that there are many false positive predictions? An initial definition of the terms would help me.
Author’s response:
The discussion paragraph in lines 368-381 has been revised to better clarify the clinical meaning and perspectives of our results (see also answer to the Reviewer’s previous comment).
Please see definitions in Statistics | Sensitivity, Specificity, PPV and NPV | Geeky Medics. We decided to not add the definitions in the revised manuscript to avoid any lengthening, but if the Reviewer wishes, we will add them.
The SEFI®-AP diagnosis of malnutrition is only considered reliable if the phenotypic criterion is the calf circumference. Here I did not understand why the BIA measurement should be less reliable as a reverse conclusion, only because the results of the SEFI®-AP would be different when using the BIA measurement as phenotypic criterion.
Author’s response:
The Reviewer is perfectly right. These are independent aspects. BIA remains a reference method for the diagnosis of malnutrition, as calf circumference is. As stated in lines 448-450, “our study suggests that the measurement of muscle mass by BIA is not performing better than the calf circumference measurement to diagnose malnutrition or sarcopenia. This result seems particularly relevant in the setting of nursing home where calf circumference is easier to implement than BIA.” But this result is independent from the fact that SEFI®-AP performs better to detect malnutrition if the chosen phenotypic criterion is the decreased calf circumference. Therefore we could not recommend to prefer calf circumference or BIA as a phenotypic criterion for malnutrition, but could conclude that “one day SEFI®-AP correctly predicts the diagnosis of malnutrition.” (lines 489-490).
Please discuss if prediction is possible if you identified that consume is stable in the setting.
Author’s response:
The Reviewer is perfectly right. The words “prediction” and “predictor” are not the right ones and were changed into “anticipation” and “marker”. The abstract has been revised as follows: “Day 3 SEFI®-AP anticipated…” (line 26) and ‘One day SEFI®-AP correctly anticipates the food intake during the following month and predicts the diagnosis of malnutrition’(lines 31-32).
The results and conclusion sections have been revised as follows:
“Day 3 SEFI®-AP <7 or ≥ 7 was considered to be a good marker if SEFI®-AP <7 or ≥7 was maintained for at least 70% of time during the corresponding days. The sensitivity of day 3 SEFI®-AP to anticipate the food intake assessment during the different periods was better for lunch and dinner than for lunch alone (Table 4).” (lines 275-278 & Table 4 legend).
“One day SEFI®-AP correctly anticipates the food intake during the following month and correctly predicts the diagnosis of malnutrition. Any decrease in food intake should lead to suspect malnutrition.” (lines 489-491).
Information on informed consent is strange. You argue that no personal contact required informed consent but what about conducting the Katz, the body weight and height etc for the GLIM, or the calf circumference and hand grip strength? Albumin level? This is part of the routine care? Please provide further details of this nursing home doing this kind of comprehensive assessment! This is unbelievable!
Author’s response:
We thank the Reviewer fir this comment. Indeed, all these parameters are integrated in the routine care in our nursing home. Moreover, in France, the French Health Authority has published in 2021 good clinical practice recommendations for the malnutrition diagnosis integrating all the nutritional parameters used in our study. To make it clearer, we have revised the ‘Ethical considerations’ paragraph of the ‘methods’ section (lines 184-185) and in the ‘Informed Consent Statement’ section (lines 507-508), as follows:
“…and the assessment of the nutritional status is part of the recommended routine care in France.”
Reviewer 2 Report
Thank you for the opportunity to review of the article titled "The Assessment of Food Intake Assisted by Photography in the Older People Living in a Nursing Home: Maintenance Over Time and Performance for the Diagnosis of Malnutrition". This study aimed to assess the maintenance over one month of one-day semi-quantitative assessment of food intake using SEFI-AP (primary aim) and its performance in diagnosing malnutrition (secondary aim) in older people living in a nursing home. In generally, nursing homes have insufficient care staff, it would be meaningful to assess food intake through photography and AI. However, some concerns need to be addressed.
1. Abstract, lines 23-24: Please provide specific values for SEFI-AP medians of lunch and means of lunch and dinner.
2. Abstract, lines 30-31: Do you mean to assessment of daily dietary intake predicts new occurrence of undernutrition one month later? When was the assessment of malnutrition? (new occurrence after one month?) This should be clearly stated in the method of the text.
3. This study proposes a new cutoff value for diagnosing malnutrition of GLIM criteria, which is the dietary intake assessed by SEFI-AP (i.e., SEFI-AP<7/10). Were the meals provided to the subject individualized to his or her individual needs? This is an essential perspective from which to consider cutoffs. For example, this cutoff means different things if a person is given too much food for his/her needs than if he/she is given inadequate food. Thus, the authors need to note in the method how the amount of food provided to each individual subject was determined. In addition, it should be shown the results that whether the SEFI-AP assessment of dietary intake reflected the energy and nutrients sufficiency in the individuals.
4. The GLIM criteria including the assessment of reduced food intake. Thus, is it reasonable in a research design to evaluate the predictive ability of the SEFI-AP food intake assessment for malnutrition which diagnosed by GLIM criteria?
5. Discussion, page 12, lines 334-335: Authors state "The performance of the one-day SEFI® to identify decreased food intake during the following month is correct and better if considering lunch and dinner rather than lunch alone." However, breakfast was not evaluated. Please state the reasons why did not assess breakfast and the limitations for this study.
Author Response
Reviewer #2:
As authors said, malnutrition in elderly is serious issue to provide better and healthy life, and, especially in nursing home, it's very hard to assess food intake correctly due to lack of time and man-power. Therefore, this paper would give us the way to assess the nutritional status with easier and less expensive way.
Author’s response:
We thank the Reviewer for her/his kind appreciation of our work.
- In nursing home, many patients have dementia or cognitive disorders, which will affect the automated assessment of food intake by photograph. In this study, the patient with severe psychotic disorder was excluded, but automated system will help the medical staff especially in these kind of patients. Therefore, it seems better for me to add the limitation of this study from the clinical point of view.
Author’s response:
The Reviewer is right. This is because patients with dementia or cognitive disorders are unable to evaluate themselves their food intake that we used the photography assistance. Only one patient with severe psychotic disorder was excluded, so this may have not biased the results. This limitation has been added in lines 466-467.
Reviewer 3 Report
As authors said, malnutrition in elderly is serious issue to provide better and healthy life, and, especially in nursing home, it's very hard to assess food intake correctly due to lack of time and man-power. Therefore, this paper would give us the way to assess the nutritional status with easier and less expensive way.
1. In nursing home, many patients have dementia or cognitive disorders, which will affect the automated assessment of food intake by photograph. In this study, the patient with severe psychotic disorder was excluded, but automated system will help the medical staff especially in these kind of patients. Therefore, it seems better for me to add the limitation of this study from the clinical point of view.
Author Response
Reviewer #3:
Thank you for the opportunity to review of the article titled "The Assessment of Food Intake Assisted by Photography in the Older People Living in a Nursing Home: Maintenance Over Time and Performance for the Diagnosis of Malnutrition". This study aimed to assess the maintenance over one month of one-day semi-quantitative assessment of food intake using SEFI-AP (primary aim) and its performance in diagnosing malnutrition (secondary aim) in older people living in a nursing home. In generally, nursing homes have insufficient care staff, it would be meaningful to assess food intake through photography and AI. However, some concerns need to be addressed.
Author’s response:
We thank the Reviewer for this comment.
- Abstract, lines 23-24: Please provide specific values for SEFI-AP medians of lunch and means of lunch and dinner.
Author’s response:
It will take too much room to provide all these specific values (see Figure 3). Indeed, as stated in the ‘statistical analyzes’ section, « a generalized linear model on ranks with repeated measures evaluated the evolution of SEFI®-AP » (lines 197-198). We have revised the abstract (lines 24-25) accordingly : « According to a generalized linear model on ranks with repeated measures, the SEFI®-AP medians of lunch (p=0.11) and means of lunch and dinner (p=0.15) did not vary over time. »
- Abstract, lines 30-31: Do you mean to assessment of daily dietary intake predicts new occurrence of undernutrition one month later? When was the assessment of malnutrition? (new occurrence after one month?) This should be clearly stated in the method of the text.
Author’s response:
We do not mean that the assessment of daily dietary intake predicts new occurrence of malnutrition one month later. The nutritional status was assessed in each patient during the first week of food intake monitoring according to the GLIM criteria.
The ‘abstract’ and ‘methods’ sections have been revised as follows:
« The nutritional status was assessed in each patient during the first week of food intake monitoring according to the GLIM criteria. » (lines 20-22 & 131-132).
- This study proposes a new cutoff value for diagnosing malnutrition of GLIM criteria, which is the dietary intake assessed by SEFI-AP (i.e., SEFI-AP<7/10). Were the meals provided to the subject individualized to his or her individual needs? This is an essential perspective from which to consider cutoffs. For example, this cutoff means different things if a person is given too much food for his/her needs than if he/she is given inadequate food. Thus, the authors need to note in the method how the amount of food provided to each individual subject was determined. In addition, it should be shown the results that whether the SEFI-AP assessment of dietary intake reflected the energy and nutrients sufficiency in the individuals.
Author’s response:
In our nursing home, the meals provided are individualized to his or her individual needs by a dedicated dietitian. The menu cycles are for 3 weeks. This has been added in the ‘methods’ section (lines 103-104). Thus, considering that a SEFI®-AP of 10 means that patients are eating according to their needs, we could consider that a SEFI®-AP<7/10 reflects insufficient food intake.
- The GLIM criteria including the assessment of reduced food intake. Thus, is it reasonable in a research design to evaluate the predictive ability of the SEFI-AP food intake assessment for malnutrition which diagnosed by GLIM criteria?
Author’s response:
In our nursing home, the patients are highly dependent with a high proportion of severe cognitive disorders (see ‘discussion’ section in lines 461-464). Thus, as mentioned in lines 155-156, “the etiologic criterion (according to GLIM) was the presence of at least one chronic disease in all the patients.” The food intake was not used as a GLIM etiologic criterion.
- Discussion, page 12, lines 334-335: Authors state "The performance of the one-day SEFI® to identify decreased food intake during the following month is correct and better if considering lunch and dinner rather than lunch alone." However, breakfast was not evaluated. Please state the reasons why did not assess breakfast and the limitations for this study.
Author’s response:
We have chosen to not include breakfast for two reasons:
- this is usually the most consumed meal of the day so not discriminative to identify patients at risk for malnutrition and worse outcomes contrary to lunch or dinner (Hiesmayr et al, ref [8])
- because of logistical reasons, it was not possible to have an investigator to monitor food intake for three meals. So based on [8], we decided to focus on lunch and dinner.
As also recommended by the Reviewer 1, we have added this information in the ’methods’ section (lines 106-110). The ‘study limitations’ section has been revised accordingly: “One limitation is that breakfast was not included, but this may have not biased the results [8] (lines 457-458).”
Round 2
Reviewer 2 Report
All my comments have been addressed